# Metabolomic Aspects of Conservative and Resistance-Related Elements of Response to *Fusarium culmorum* in the Grass Family

**DOI:** 10.3390/cells11203213

**Published:** 2022-10-13

**Authors:** Anna Piasecka, Aneta Sawikowska, Natalia Witaszak, Agnieszka Waśkiewicz, Marta Kańczurzewska, Joanna Kaczmarek, Justyna Lalak-Kańczugowska

**Affiliations:** 1Institute of Bioorganic Chemistry, Polish Academy of Sciences, 61-704 Poznań, Poland; 2Department of Mathematical and Statistical Methods, Poznań University of Life Sciences, 60-637 Poznań, Poland; 3Institute of Plant Genetics, Polish Academy of Sciences, 60-479 Poznań, Poland; 4Department of Chemistry, Poznan University of Life Sciences, 60-637 Poznań, Poland; 5Institute of Mathematics, Poznan University of Technology, 60-965 Poznań, Poland

**Keywords:** FHB, plant metabolomic, plant–pathogen interaction, barley, wheat, *Brachypodium distachyon*, pathway enrichment

## Abstract

Background: *Fusarium* head blight (FHB) is a serious fungal disease affecting crop plants, causing substantial yield reductions and the production of mycotoxins in the infected grains. Achieving progress in the breeding of crops with increased resistance and maintaining a high yield is not possible without a thorough examination of the molecular basis of plant immunity responses. Methods: LC-MS-based metabolomics approaches powered by three-way ANOVA and the selec-tion of differentially accumulated metabolites (DAMs) were used for studying plant immunity. A correlation network and functional enrichment analysis were conducted on grains of barley and wheat genotypes that were resistant or susceptible to FHB, as well as on the model grass *Brachypodium distachyon* (Bd), as this is still poorly understood at the metabolomic level. Results: We selected common and genotype-specific DAMs in response to *F. culmorum* inoculation. The immunological reaction at the metabolomic level was strongly diversified between resistant and susceptible genotypes. DAMs that were common to all tested species from the porphyrin, flavonoid, and phenylpropanoid metabolic pathways were highly correlated, reflecting con-servativeness in the FHB response in the *Poaceae* family. Resistance-related DAMs belonged to different structural classes, including tryptophan-derived metabolites, pyrimidines, the amino acids proline and serine, as well as phenylpropanoids and flavonoids. The physiological re-sponse to *F. culmorum* of Bd was close to that of barley and wheat genotypes; however, metabo-lomic changes were strongly diversified. Conclusions: Combined targeted and untargeted metabolomics provides comprehensive knowledge about significant elements of plant immuni-ty that have the potential to be molecular biomarkers of enhanced resistance to FHB in the grass family. Thorough examination of the Bd metabolome in juxtaposition with diversified geno-types of barley and wheat facilitated its use as a model grass for plant–microbe interaction.

## 1. Introduction

Cereal crops in Europe are dominated by common wheat (44% of yield), as well as a steadily growing proportion represented by barley (18.6%), clearly ahead of grain maize (23.4%) (data of Eurostat, https://ec.europa.eu/eurostat (accessed on 1 July 2022)). Nevertheless, the resulting cereal yield reaches approximately half of the genetic potential, and one of the main reason for yield reduction are diseases caused by pathogenic fungi, in particular from the toxinogenic genus *Fusarium* [1]. The most common symptoms of *Fusarium* wilt in cereal are instant *Fusarium* head blight (FHB), root rot, and *Fusarium* seedling blight (FSB). *Fusarium* head blight caused by number of pathogenic *Fusarium* species, including *Fusarium culmorum,* is a serious fungal disease of small cereal grains such as wheat, barley, and oats, as well as forage grasses [2]. The disease develops from infection at the anthesis and spreads until grain harvest resulting in handicapped kernel development. The main problem caused by the FHB is grain contamination with mycotoxins, such as zearalenone (ZEN), fusarins, and type B trichothecenes, as well as deoxynivalenol (DON), which is considered to be the most toxic compound [3]. There are no cereal varieties that are fully immune to FHB, as it is a quantitative trait; therefore, efficient crop breeding must be strongly supported by comprehensive knowledge regarding the molecular mechanisms of pathogenesis and plant–-fungus interactions. One of the basic factors influencing the resistance of cereals is their composition of low molecular metabolites and the speed of their activation and biosynthesis.

Although there are many immunological similarities among genetically close species of the Poaceae family, there is a high level of diversification in the metabolite profiles in this taxonomic group. For example, benzoxazinoids are detected in wheat, rye, and maize, whereas barley and *Brachypodium distachyon* (Bd) display the loss of the biosynthetic genes of benzoxazinoids [4]. Cereals’ responses to FHB in the context of conserved mechanisms should therefore be examined with consideration of the existence of cultivars with distinct levels of FHB resistance. Various defensive metabolites are derived via the phenylpropanoid ⁄ shikimic acid pathway in *Poaceae* plants. The accumulation of hydroxycinnamic acid amides and phenolic acid glycosides in wheat’s response to FHB was correlated with cell wall thickening and the inhibition of pathogen growth [3]. The key enzymes of phenylpropanoid biosynthesis, such as cinnamyl alcohol dehydrogenase, caffeoyl-CoA *O*-methyltransferase, caffeic acid *O*-methyltransferase, flavonoid *O*-methyltransferase, agmatine coumaroyltransferase, and peroxidase, were up-regulated in response to *F. graminearum* and DON production [5]. Several metabolomics studies have investigated flavonoids that were correlated to FHB resistance in barley [6,7] and wheat [8]. Among these, the glycosidic forms of the flavonols kaempferol and quercetin have been observed as a highly abundant group which can inhibit fungal transcription [9]. A deficiency in proanthocyanidin production facilitated the increased penetration of the barley seed testa layer by both *F. graminearum* and *F. culmorum* [10].

Recently, much attention has been paid to antifungal properties of polyamines. Putrescine, spermidine and spermine were noted as rapidly induced in FHB and contributed to the induction of DON biosynthesis by *F. graminearum* [11]. Polyamines acylated with hydroxycinnamic acids accumulate during pathogenesis in barley [6,7], wheat [11] and the model cereal plant *Brachypodium distachyon* [5].

*Brachypodium distachyon* (Bd) is used as a model system for analyses of the functional genomics of grasses. In recent years, Bd has emerged as a model plant for studies of cereal crop–pathogen interactions [12]. To date, however, there has been very little evidence to demonstrate the conservation of function between *B. distachyon* and cereals with respect to immunological processes at the molecular level. Many biological tests and histopathological studies have shown that *Fusarium* spp. infect Bd in a similar manner to the way in which they infect wheat [12], but a positive correlation between Bd and barley physiology was also demonstrated. To enhance our knowledge of Bd as a model grass, we compared the most commonly used Bd21 line, which is still poorly known at the metabolomic level, to well-characterized genotypes of wheat and barley with widely differing susceptibility to FHB symptoms. Therefore, we presented conservative and genotype-specific metabolites in terms of their susceptibility and resistance to FHB in the expanded context of the *Poaceae* family. Distinguishing between conservative and resistance-related defense elements can also facilitate strategies for plant breeding in cereal crops with enhanced resistance to FHB.

## 2. Materials and Methods

### 2.1. Reagents and Chemicals

Acetonitrile for LC-MS analyses was obtained from VWR Chemicals (Radnor, PA, USA), formic acid was obtained from Merck SA (Darmstadt, Germany). Ultrapure water was obtained from a Millipore Direct Q3 device (Merck SA). Standards of compounds (ferulic acid, caffeic acid, phenylalanine, catechin, apigenin, quercetin, isoorientin, apigenin 6-*C*-glucoside-8-*C*-arabinoside, apigenin 6,8-di-*C*-glucoside, 4-hydroxybenzoic acid, luteolin-3,7-di-*O*-glucoside, tricin 7*-O*-glucoside, abscisic acid) were purchased from Extrasynthese (Genay, France). Isoorientin 2”-*O-*glucoside, isovitexin 7-*O*-glucoside, isoscoparin 2”-*O*-glucoside, and apigenin 6-*C-*arabinoside-8-*C-*glucoside were purified from plant material and their structures were confirmed with NMR analysis as described previously [13]. Standards of zearalenone, tryptophan, 2,2-diphenyl-1-picrylhydrazyl, and ascorbic acid were obtained from Merck SA. Hypochloride was obtained from BioShop (Burlington, ON, Canada).

### 2.2. Plant Growth and Inoculation

Two spring wheat (*Triticum aestivum*) and barley (*Hordeum vulgare*) genotypes, differing in their susceptibility to *F. culmorum* and, in addition, the model plant *Brachypodium distachyon* line Bd21 (Bd21), were selected for experiments. For simplicity, genotypes with high susceptibility to FHB are hereafter referred to as susceptible, whereas genotypes with reduced susceptibility to FHB will be referred to as resistant, although in fact their resistance is not complete.

Plants were cultivated in the phytotron of the Plant Cultivation Center of the Institute of Plant Genetics, Polish Academy of Sciences, Poland, Poznań. Barley genotypes came from recombinant inbred lines of *Hordeum vulgare* MCam, obtained from the cross between the Polish cultivar Lubuski and a Syrian breeding line—Cam/B1/CI08887//CI05761—selected from the materials of the Institute of Plant Genetics, Polish Academy of Sciences on the basis of previous studies [14]. Two extremely different lines of barley—the MCam 88 line (hereafter referred to as Hs) and the resistant MCam 59 line (hereafter referred to as Hr)—were taken for metabolomic analysis. The sensitive cultivar Radocha [15] (hereafter referred to as Ts) and the highly resistant line CJ 9306 [16] (hereafter referred to as Tr) of wheat were obtained from Breeding Company Strzelce, Poland. The Bd21 line was provided by the National Plant Germplasm System of USDA-ARS.

Seeds were surface-sterilized by soaking them in 2.5% sodium hypochlorite for 30 min, then washed thoroughly in sterile deionized water. The seeds were sown into pots of 30 cm in diameter filled with soil in growth chambers. During the first month, plants were grown under short-day conditions (10 h light, 14 h dark) at 16 °C and then switched to long-day conditions (16 h light, 8 h dark) at a temperature of 21 °C in the light/14 °C in the dark with a humidity of 60% of RH for the acceleration of flowering. Plants were grown in two principal experiments.

An inoculation was performed with the fungus *F. culmorum* isolate KF846 from the collection of the Institute of Plant Genetics, Polish Academy of Sciences. To better characterize the *F. culmorum* strains, the expression of key biosynthetic genes of mycotoxins (zearalenone synthase—*ZEA2*; zinc finger transcription factor—*TRI6*; trichodiene synthase—*TRI5*) were performed, leading to the characterization of the mycotoxinogenic profile of this strain (see Appendix A). The *Fusarium culmorum* isolate KF846 produces mycotoxins from the group of trichothecenes and estrogens. Genes related to the biosynthesis of estrogenic zearalenone show much higher expression than genes related to the biosynthesis of trichothecenes. In order to prepare the spore suspension, spores were washed from the surface of a 2-week-old colony growing on PDA medium (Potato, Dextrose, Agar, Oxoid). The inoculation was carried out at the concentration of 10^6^ spores/mL.

The inoculation with fungal spores was carried out in the flowering stage of the spikes (at stage 61 of the BBCH scale) by spraying. For barley and wheat, this was carried out at approx. 8 weeks, and for Bd21, 12 weeks from sowing. After inoculation for 24 h, the plants were kept in the dark with mist irrigation to increase the efficiency of infection. For gene expression analysis, samples (spikes) were collected 24 and 48 h after inoculation. For metabolomic analysis, samples were collected 4 and 6 days after inoculation. For antioxidant and mycotoxin analysis, samples were collected 6 days after inoculation. All collected samples were immediately frozen in liquid nitrogen and stored in −80 °C until assays.

### 2.3. Quantification of Fusarium Culmorum

DNA was extracted using CTAB (cetyltrimethyl ammonium bromide). The samples were suspended in 650 μL CTAB and incubated at 65 °C for 20 min. A volume of 500 μL CHCl_3_ was added and centrifuged at 12,879× *g* for 15 min. DNA was precipitated with 65 μL 3M sodium acetate, pH 5.4, and two volumes of ice cold 99.8% ethanol. The tubes were stored at −20 °C overnight and centrifuged at 17,530× *g* for 5 min. The pellets were washed with 70% ethanol, centrifuged at 17,530× *g* for 5 min and fully dissolved in 100 μL of TE buffer.

Real-time PCR was performed in 10 μL containing 7.5 μL AmpliQ Real-Time PCR Opti Probe Kit (Novazym, Poznań, Poland), 100 nM of FAM-labeled probe, and 300 nM of forward and reverse *F. culmorum* primers (see Appendix A) [17]. The thermal cycling parameters for quantitative fungal DNA detection were 95 °C for 2 min, followed by 40 cycles of 95 °C for 15 s and 60 °C for 30 s. Nuclease-free water was used as the no-template control. A standard curve was generated by plotting the Ct value for each sample of a standard series of the amount of fungal biomass (10–0.001 mg/g). All the samples were tested in triplicate.

### 2.4. Zearalenone Measurements

Organic solvents (HPLC-grade) for analysis were purchased from Sigma-Aldrich (Steinheim, Germany). HPLC-grade water was obtained using a Milli-Q system (Millipore, Bedford, MA, USA). The stock solution of zearalenone (ZEN) was dissolved in methanol (1 mg/mL) and stored at −20 °C. ZEN was extracted and purified on a ZearalaTest column (Vicam, Milford, CT, USA) according to a procedure described previously [18]. The chromatographic elution of ZEN was performed on a Waters 2695 HPLC (Waters, Milford, CT, USA) system with a Waters 2996 Photodiode Array Detector and a Nova Pak C-18 column (150 × 3.9 mm). The data were processed using the Empower software version 1 (Waters, Milford, CT, USA). Quantitative estimations of ZEN were performed by measuring the peak areas at the retention time according to the relative calibration curve. The limit of detection was 0.01 μg/g.

### 2.5. Antioxidative Activity Measurements

The antioxidative state was measured by means of the 2,2-diphenyl-1-picrylhydrazyl (DPPH) radical scavenging activity assay, according to the modified method presented in [19] with ascorbic acid as a standard. The DPPH-scavenging activity of methanolic extracts was measured via incubation with 200 µL of 0.2 mM DPPH radicals for 40 min and the absorbance (A) was measured at 515 nm. The percentage of radical scavenging (%Quenching) was calculated using the following formula: Q(%) = (Ablank-Asample)/Ablank × 100.

### 2.6. LC-MS Analysis

Methanolic extracts for LC-MS analysis were prepared according to [20]. The LC-MS system consisted of UPLC with a photodiode-array detector PDAeλ (Acquity System; Waters) hyphenated to a high-resolution Q-Exactive hybrid MS/MS quadrupole Orbitrap mass spectrometer (Thermo Fisher Scientific, Waltham, MA, USA). Chromatographic profiles of metabolites and quantitative measurements were obtained using water acidified with 0.1% formic acid (solvent A) and acetonitrile (solvent B) with a mobile phase flow of 0.35 mL/min on an ACQUITY UPLC HSS T3 C18 column (2.1 × 50 mm, 1.8 μm particle size; Waters) at 22 °C. The injection volume was 5 μL.

Q-Exactive MS operated in Xcalibur version 3.0.63 with the following settings: heated electrospray ionization ion source voltage −3 kV or 3 kV; sheath gas flow 30 L/min; auxiliary gas flow 13 L/min; ion source capillary temperature 250 °C; auxiliary gas heater temperature 380 °C. MS/MS mode (data-dependent acquisition) was recorded in negative and positive ionization, at a resolution of 70,000 and the AGC (ion population) target 3 × 10^6^, and a scan range of 80 to 1000 m/z.

The obtained LC-MS data were processed for peak detection, deisotoping, alignment and gap filling using MZmine 2.51 [21] separately for positive and negative ionization mode; then, data from both modes were combined. Signals corresponding to mycotoxins identified in LC-MS/MS on the basis of literature data, databases, and fragmentation spectra were removed from analysis (Appendix A). The prepared data table was post-processed for missing value imputation, log transformation, and data filtering for further statistical analysis. Processed datasets from positive and negative ionization were deposited in a publicly available database, Mendeley Data, with the DOI: 10.17632/27gcwgkfnj.1 while raw data in universal mzML format is available at https://box.pionier.net.pl/d/38d066ad1dc54503adfe/ (accessed on 3 October 2022).

Global LC-MS analysis enabled us to detect five mycotoxins: zearalenone (ZEN), nivalenol (NIV), deoxynivalenol (DON), diacetoxyscirpenol (DAS),) and the T2 toxin, on the basis of *m/z* values and fragmentation spectra, according to [22]. Type A-trichothecenes, including DAS and T2 toxin, were detected in positive ionization mode as ammonium adduct [DAS+NH_4_]^+^ and [T2+NH_4_]^+^ ions at *m*/*z* = 384.2022 Da and 484.2546 Da, respectively (Appendix A). For type-B trichothecenes NIV and DON, the respective precursor ions [NIV+CH_3_COO]^−^ and [DON+CH_3_COO]^−^ represented adducts with formic acid at *m/z* =371.1342 Da and *m/z* =355.1392, respectively. Estrogenic zearalenone detection was conducted on the basis of [M-H]^-^ ions at *m/z* = 317.1472624 Da. These signals were removed from the data table providing statistics for the visualization of plant immunity responses (described in Section 3.4.).

### 2.7. Statistical Analysis

The processed LC-MS dataset consisted of 5 experimental groups (genotypes) in 2 treatments (control and infection), 2 time points (T1,T2), 4 biological replications, and 2 experiments, with 16,781 signals for negative ionization and 8132 signals for positive ionization. Observations equal to zero (below the detection level) were substituted by half of the minimum non-zero observations for each signal. Then observations were transformed via logarithmization in order to bring the data distribution close to a normal distribution. Three-way analysis of variance (ANOVA) was performed with experiments as a block (random effects) and treatment, genotype, and time point as 3 fixed factors. Statistical analysis was performed together for positive and negative ionization. The resulting p-values were corrected for multiple testing by calculating q-values (the false discovery rate, FDR). Significant changes in the accumulation of metabolites were indicated for the effect of treatment, for the effect of genotype, for the effect of time points, and all their possible interactions, with q-values < 0.05 (Appendix A).

Differentially accumulated metabolites (DAMs) between inoculated and control plants for each species in each time point were selected with the following conditions: if any effect containing treatment was significant (treatment or interaction treatment × species or interaction treatment × time point or interaction treatment × species × time point) with q-values < 0.05 and |log2FC| > 0.58, where FC is the fold-change relative to treatment/control for each species in each time point. The tables containing results of the ANOVA and the DAM selection process are available in Appendix A.

ANOVA was performed in Genstat 21 (VSN International, Hempstead, UK). The false discovery rate was calculated in R 4.0.4. Visualizations, including PCA, heatmaps, Venn diagrams, dendrograms and barplots, were created in R 4.0.4. PCA was carried out for each time point on the data after logarithmic transformation and the fold-change calculation (log2FC) only for DAMs. Venn diagrams for different time points were constructed based on the DAMs for each genotype. Dendrograms were constructed based on means in 4 variants, for controls at T1, for controls at T2, for the infection group at T1, and for the infection group at T2. Barplots were constructed based on the number of DAMs for each genotype and time point. Heatmaps showing differences between treatments and controls for each time point and for each genotype were constructed based on log2FC values (sorted firstly in descending order by Hr and Hs, then Tr and Ts, then secondly reordered based on row means) for selected annotated metabolites meeting the condition for the definition of DAMs common in resistant genotypes of Tr and Hr but not being DAM in other genotypes or common in susceptible genotypes Ts and Hs but not being DAM in other genotypes for both time points separately.

Four correlation networks were constructed separately for each time point in the control and treatment groups using the WGCNA package in R [23] and they were visualized using Cytoscape [24]. First, the Pearson correlation matrix was transformed into an adjacency matrix using a power function of 6 in all cases according to the scale-free topology criterion [23]. Modules were detected via clustering, using the dynamic tree cut algorithm on the topological overlap matrix, which was visualized via Cytoscape. For all networks, hubs, defined as highly connected metabolites, were selected as nodes with the highest number of connections.

### 2.8. Functional Analysis and Characterization of Metabolites

Signals selected as DAMs were combined from positive and negative ionization results and were imported into the Functional Analysis module in MetaboAnalyst 5.0 [25] as a peak list profile. Annotation was performed with a 5 ppm mass tolerance and the Mummichog algorithm with 0.05 p-value cutoff on the basis of the *Oryza sativa japonica* (Japanese rice) reference metabolome in the Kyoto Encyclopedia of Genes and Genomes (KEGG) [26]. The annotated compounds were subsequently subjected to the Pathway Analysis and Enrichment Analysis modules in MetaboAnalyst 5.0. Metabolites over-represented at the pathway level were ranked on the basis of hypergeometric testing, followed by Benjamini–Hochberg false discovery rate (FDR) correction. The relative importance of individual nodes to the overall pathway network (pathway topology) was scored on the basis of relative-betweenness centrality. Significantly enriched metabolic pathways upon differentiating factors were selected if FDR was <0.03 consistently across multiple comparisons, as presented in Table 1. All the tables of results, containing the pathway mapping scores for functional annotation and enrichment impact are available in Appendix A for each analysis, with two sheets. The sheet entitled “pathway enrichment” contains the scoring of the enrichment of the entire pathway for all experimental groups. The sheet entitled “annotation” contains information about the annotated metabolites from the KEGG [26], HMDB [27], and PubChem [28] databases. Metabolites that were over-represented according to the structural classification were ranked on the basis of metabolite set enrichment analysis (MSEA) grouping with FDR < 0.01. The results regarding significant enrichment based on structural classification are also presented in Table 1 and a complete table of results containing the scoring of structural mapping is available in Appendix A.

The annotation of individual DAMs carried out through enrichment analysis was complemented by tentative DAM identification on the basis of *m/z* values and fragmentation in MS/MS spectra, and confirmed via comparison of the exact molecular masses with ∆ of less than 5 ppm—in terms of their retention times and mass spectra—to those of standard compounds or spectra in the available databases (PubChem [28], KNApSAck [29], ChEBI [30], Metlin [31], ReSpect [32]) and using literature data. The confirmation of the aglycone type of flavonoid isomers (e.g., kaempferol and luteolin differentiation) was based on the comparison of standard compounds and the use of a method that has been described previously [33].

## 3. Results

### 3.1. Progress of F. culmorum Infection Was Diversified among Genotypes

The genotypes of barley and wheat that were characterized as susceptible to FHB had a higher level of fungal biomass than resistant ones. It should be noted that, in Ts, the biomass of *F. culmorum* was much more expanded than that in Hs (Figure 1A). The fungal biomass in Bd was at a similar level as that observed for Hs and was more expansive than that observed in resistant genotypes. Differences in biomass production between both wheat cultivars were prominent, with statistical significance *p* < 0.05, whereas both barley genotypes manifested moderate differences.

### 3.2. Antioxidant Capacity Was Triggered in Susceptible but Not in Resistant Genotypes

The total antioxidative activity was evaluated using in vitro methanolic extracts of the studied plants. In resistant genotypes, the antioxidant capacity was similar in control and infected plants (Figure 1B). In Hs, the antioxidant capacity decreased slightly, whereas in Ts and Bd the decrease was significant. Interestingly, in control plants, the antioxidant capacity of Tr was constitutively lower than those in other studied grasses.

### 3.3. Production of Pathogen-Derived Mycotoxins was Impaired in Resistant genotypes of Poaceae 

Genetic characterization of the *F. culmorum* strain revealed the expression of genes related to the production of mycotoxins from trichotecenes and estrogen classes (Appendix A). The in vitro assay showed a relatively higher level of gene expression for estrogenic zearalenone (zearalenone synthase—ZEA2) than trichotecenes (zinc finger transcription factor—*TRI6*, trichodiene synthase—*TRI5*). Processed and normalized data enabled the comparison of the accumulation of detected mycotoxins. Diacetoxyscirpenol was significantly accumulated in both barley genotypes, whereas nivalenol was more pervasive in wheat genotypes (Figure 1C–G). The level of T2 toxin was close to the detection limit in resistant genotypes Tr and Hr. Susceptible genotypes Ts and Hs had high levels of T2 toxin accumulation, and the highest level was displayed by Bd21. Interestingly, the trend of accumulation for other detected mycotoxins in Bd21 was similar to those of the susceptible genotypes, except for zearalenone, of which the trend of accumulation was close to that of the resistant genotypes. The time point of mycotoxin measurement had a weak effect on the accumulation of mycotoxins and the level was stable from T1 to T2.

Processed LC-MS data cannot provide quantitative results; therefore, the most highly accumulated mycotoxin of the *F. culmorum* strain used in our experiments, zearalenone, was quantitatively measured in spikes on the basis of a UV calibration curve. The quantitative measurement of ZEN showed the same kinetic of accumulation as was detected by LC-MS. Resistant variants of barley and wheat were characterized by significantly lower levels of ZEN (less than 10 ng/g of dry weight) in comparison to susceptible plants (Figure 1H). The results obtained for Bd (14 ng/g of dry weight) were close to the results for the resistant variants. Large discrepancies in the results for susceptible genotypes were observed. The highest level of zearalenone was detected in Ts (about 150 ng/g of dry weight), and it was almost 10 times more prevalent than the level observed in Tr. It is worth mentioning that the European Union norm for zearalenone accumulation in spikes is 100 ng/g (100 ppb).

### 3.4. Conserved DAMs Were Highly Correlated, Whereas Genotype-Specific DAMs Determined Variation in Resistance to F. culmorum

Three-way ANOVA gave realistic significance of signals for all three factors: genotype, treatment, and time point, as well as their interactions. The highest number, 19,596 of significant signals was for genotype factor followed by interaction time point × genotype with—9941 significant signals (Appendix A).

The calculation of the fold-change between inoculated and control plants, complimented by the selection of statistically significant signals via ANOVA, enabled us to select differentially accumulated metabolites (DAMs). It followed by indication of metabolites that were common to all genotypes and genotype-specific in the immune responses of *Poaceae* plants (Appendix A). A strong metabolomic response to *F. culmorum* inoculation was observed in all studied genotypes since thousands of signals had changed their accumulations, compared between control and inoculated groups (Figure 2A). Even at an early infection stage, at T1, the number of signals with increased or decreased accumulation was more than two thousand for every genotype. At T2, the number of DAMs significantly increased for every genotype. Proportionally fewer DAMs—both with decreased as well as increased accumulation—were observed for barley genotypes, whereas wheat and Bd had extended numbers of DAMs. In wheat and Bd, the difference in the number of DAMs when comparing T1 and T2 was more evident, especially for DAMs with decreased accumulation. The lowest number of DAMs with the least variation from T1 to T2 was observed in barley genotypes. The metabolomic profile of the Bd21 control plants was close to that of wheat control plants, especially in regard to resistant Tr, whereas both barley genotypes were different from each other and the rest of the studied genotypes (Figure 2B). The reprogramming due to the response to *F. culmorum* caused the approximation of the metabolome profiles between Bd21 and barley genotypes at both time points (Figure 2C). At T2, the metabolomic profiles of all plant genotypes shifted to higher inter-species distances.

Overall, the metabolomic immune response to *F. culmorum* was diversified among all the tested genotypes (Figure 2D). The DAMs of Bd21 showed the highest distance from other groups at both time points. The DAMs of barley and wheat showed more intra-species similarities from T1 to T2. In spite of the differences in their metabolomic profiles, the conservatism of the responses among all tested genotypes was significant. The number of common DAMs observed at a later stage of infection—at T2 (1307 common signals)—was greater than that observed at T1 (839 common signals) (Figure 2E). Elements that were specific only for susceptible or resistant genotypes could also be observed. Tr and Hr had 136 shared DAMs at T1. This number decreased to 47 at T2. On the other hand, Ts and Hs had 165 shared DAMs at T1 and this number increased at T2 to 265. Bd21 had the highest number of shared DAMs with wheat genotypes. At T1 there were 309 shared DAMs between Bd21 and Tr, whereas at T2 a higher commonality was shown with Ts (304). Both barley genotypes shared a significantly lower number of DAMs than wheat genotypes. There were 85 and 73 DAMs shared for barley genotypes at T1 and T2, respectively, in comparison to 664 and 287 DAMs at T1 and T2, respectively, for wheat genotypes.

Due to the large number of common DAMs, presented in the Venn diagram (Figure 2E, 839 at T1 and 1307 at T2), we checked the correlation networks among them in control conditions and after the progression of *F. culmorum* infection. Those signals showed a low level of connectivity at both time points for control conditions, whereas biotic stress caused an increase in the correlation between particular DAMs (Figure 3 for T2 and Appendix A for T1). Upon infection, correlation became more intricate and extended beyond modules. The most numerous turquoise module under control conditions was divided into two modules—turquoise and light cyan—after *F. culmorum* inoculation due to the enhancement of correlations among particular DAMs. The inter-modular connectivity became especially noticeable for unsaturated fatty acids and phenylpropanoids. DAMs with the highest number of connections were assigned as hubs; predominant among these were ferulic and sinapic acid and two isomers of *p*-coumaric acid glucoside, incorporated into the largest turquoise module. The highest number of hubs belonged to the purple module, in which flavonols, apigenin and tricin, ferulic acid derivative, and phosphorylated thymidine were correlated with each other. Jasmonic acid showed strong intra-modular negative correlation. However, its derivative MeJa showed a strong positive correlation within the cyan module.

### 3.5. Evolutionarily Conserved Metabolic Pathways Enriched Common DAMs

Firstly, all common DAMs taken for the correlation network, were subjected to enrichment analysis. Secondly, the annotated metabolites were grouped according to their structural classes.

“Galactose metabolism” had the highest score of enrichment at T1 and T2 (Table 1) among the common DAMs on the basis of FDR significance and pathway impact. Most of the annotated compounds belonged to module of “Raffinose family oligosaccharides“. Furthermore, photosynthesis-related “Porphyrin and chlorophyll metabolism” was enriched at both time points, and this incorporated metabolites from the “Heme biosynthesis” module. Phenylpropanoids were the most numerous annotated group of metabolites at both time points. Functional annotation on the basis of the KEGG enriched the “Phenylpropanoid biosynthesis” pathway at T1, incorporating *p*-coumaric, caffeic, ferulic and sinapic acids, and their aldehyde derivatives. At T2, spermidine conjugated with hydoxycinnamic acids and monolignols (annotated as syringin and coniferin and derivatives) were additionally matched. Dominant DAMs at T1, characterized via searches of additional databases, were derivatives of apigenin, kaempferol, quercetin, and chalcones from “Flavonoids biosynthesis”. In contrast, at T2, only one flavonoid, pentahydroxyflavanone, was identified. Pathways related to aromatic amino acids from “Tyrosine metabolism” at T1 and “Tryptophan metabolism“ at T2 were also commonly changed in grasses. The highlighting of tryptophan-derived metabolites at T2 was complimented by database annotations of indoleacetic acid and its 5-hydroxylated form, as well as tryptamine and *N*-salicyloyltryptamine. Ascorbic acid was matched as a central metabolite from the “Ascorbate and aldarate metabolism” pathway. Unsaturated fatty acids from “Arachidonic acid metabolism”, but not arachidonic acids themselves, as well as central regulatory ”2-Oxocarboxylic acids”, were triggered at T2. The diversification of the chemical structures of DAMs throughout the progression of infection from T1 to T2 reflects this structural classification. At T1, enrichment with FDR < 0.01 was observed for 16 metabolite classes, whereas at T2, this number was increased to 20 classes (Table 1 and Appendix A). Grouping annotated signals on the basis of structural classes confirmed the strong changes among benzamides, monosaccharides, isoprenoids, purines and pyrimidines, TCA acids, amino acids, and peptides and porphyrins at both time points.

### 3.6. Pathways including Four-Nitrogen-Containing Metabolites and Amino Acids Can Accelerate the Resistance of Barley Genotypes

Similar metabolic pathways were triggered at both time points in the Hs genotype. This enrichment concerned the lignan biosynthesis of matairesinol and secoisolariciresinol (in KEGG incorporated to “Biosynthesis of secondary metabolites—other antibiotics”), as well as “α-Linolenic acid metabolism” and the gibberellin biosynthesis pathway from “Diterpenoids metabolism”. Specifically at T2, DAMs from “Arachidonic acid metabolism” were matched, whereas at T1, “Ascorbate and aldarate metabolism“ was triggered.

The best score for Hr at T1 was observed for the metabolism of amino acids and derivatives from “Arginine and proline metabolism” and phenolics from “Phenylpropanoid biosynthesis”. The “2-Oxocarboxylic acid chain extension” pathway was represented at both time points by sulfur-containing dicarboxylic acids—methylthiopentylmalic acid isomers at T1 and methylthiobutylmalic acid isomers at T2. Other matched metabolites belonged to flavonoids, mainly glycosidic forms of flavonols. In further stages of infection at T2, four-nitrogen-containing metabolites were annotated, as suggested by ”Caffeine metabolism” and “Purine metabolism” enrichment. In ”Caffeine metabolism”, paraxanthine and theobromine, which were transformed to 1,7-dimethyluric acid and 3,7-dimethyluric acid, respectively, were annotated. In the indicated “Purine metabolism”, adenosine, deoxyguanosine, and allantoic acid were annotated. Interestingly, cAMP and GMP were matched at T1 and T2, respectively.

The structural over-representation of DAMs for Hs was indicated in the case of six metabolite classes at T1, with the reduction of this number to three at T2, predominantly representing saccharides and purines. Amino acids and peptides, together with benzamides, were selected at T1, whereas only purines were selected at T2 for Hr.

### 3.7. Pathways Related to Unsaturated Fatty Acids and Cell Wall Biopolymers Were Enriched in Resistant Wheat

Similar to Tr, Ts exhibited significantly altered “Arachidonic acid metabolism” during *F. culmorum* infection. The lignan pinoresinol and its derivatives detected at T1 were matched to “Biosynthesis of secondary metabolites—other antibiotics”. Enriched “Caffeine metabolism” is related to metabolites with purine rings. Mono- and di-terpene skeletons were mainly related to the “Gibberellin A12 biosynthesis” module. In Tr at T1, phenolics-related pathways were mainly enriched. Additional database searches revealed the prevalence of different hydroxycinnamic acids. An equally large class constituted flavonoids, represented by chalcones, flavones, flavanones, and flavanols. Pathways predominant in Tr at T2 incorporated unsaturated fatty acids and phospholipids. Steroidal structures of cholic acid and their conjugates were also characteristic DAMs for wheat. Those molecules, in cooperation with glycerols and phenolics, serve as precursors for “Cutin, suberine, and wax biosynthesis”, of which FHB-related changes were also noted in Tr.

### 3.8. Bd21 Induced Pathways Governing Bioenergetics in Response to F. culmorum

The DAMs enriched for Bd21 differed significantly between both time points. At T1, the annotated metabolites belonged predominantly to pathways responsible for bioenergetic supply—“Citrate cycle (TCA cycle)” and “Glycolysis/Gluconeogenesis”. Sugar metabolism pathways linked to “Amino sugar and nucleotide sugar metabolism”, “Pentose phosphate pathway”, and “Fructose and mannose metabolism” were also FHB-responsive. In addition, “α -Linolenic acid metabolism” was enriched at T1 and “Linoleic acid metabolism“ was matched at T2. The enrichment of DAMs at T2 were focused on chlorogenic acid from “Phenylpropanoids biosynthesis”, quercetin and kaempferol, flavonols from “Flavone and flavonol biosynthesis”, and uric acid derivatives from “Purine metabolism”, which were mainly strengthened by inosine- and adenine-related compounds.

### 3.9. Putative Metabolomic Biomarkers of Resistance to F. culmorum Belong to Amino Acids, Pyrimidines, Phenolics, and Jasmonic acid Derivatives

In order to check if any metabolite could be related to the enhancement of FHB resistance, we selected DAMs that were shared by both resistant genotypes, Tr and Hr (Figure 4, Appendix A). The relatively high number of DAMs shared for Tr and Hs at the beginning of infection (at T1 was 136 shared DAMs) turned into a species-specific metabolic response, since only 47 DAMs were selected at T2. Consequently, we also annotated DAMs that were common to susceptible genotypes. For T1 and T2, 165 and 265 signals (respectively) common to Ts and Hs were taken for functional analysis. Database annotation revealed large discrepancies between specific DAMs from T1 and T2, since no metabolites recurred between T1 and T2. Interestingly, database searches revealed *S*-containing metabolites, sulfanilides, *N*-acetyl-methionine, and methioninesulfoxide as shared DAMs for resistant genotypes (Figure 4). However, the amino acid methionine was detected as a DAM for susceptible genotypes. The amino acids proline, and serine, were also DAMs for resistant genotypes. Methoxylated and hydroxylated forms of indoleacetic acid were found at T1 as resistance-related DAMs. Similarly, methoxylated and hydroxylated forms of flavones, e.g., syringetin, which is 3’,5’-dimethoxy-3,5,7,4’-tetrahydroxyflavone, and tricin, which is 3’,5’-dimethoxy-4’,5,7-trihydroxyflavone were matched. Moreover, prenylation of a variety of flavonoid structures were also observed. Hydroxycinnamic acids shared in Tr and Hr were mainly present as conjugates with phenolic compounds, as in the example of rosmarinic acid and amurensin, however, with the opposite effect of accumulation, comparing between barley and wheat. The lignan secoisolariciresinol was matched with DAMs at T1 for resistant genotypes, whereas other lignans—lyoniresinol, pinoresinol, and matairesinol—were found to be DAMs in susceptible genotypes. Database searches revealed hordatine glucosides as DAMs, differentiating barley genotypes. Interestingly, signals matched to glucosides of hordatines A and B, and C were selected as DAMs, rather than hordatines alone. Apart from flavonoids and phenylpropanoids, pyrimidine and purine derivatives also constituted a numerous group of differentiated DAMs. The uridine derivative increased in Hr, but opposite changes were observed for Tr. On the other hand, the pyrimidine-2,4-diamine derivative was distinguished by its high increase in Tr in comparison to Hs. The principal phytoalexins in wheat (as well as rye and maize) are specialized metabolites belonging to hydroxamic acids (benzoxazinoids) [4]. Interestingly, only DIBOA glucoside and di-hexoside, DIMBOA glucoside, HMBOA glucoside, and HDMBOA hexoside were identified as FHB-related DAMs of wheat among the benzoxazinoids (Figure 4). However, the trends of accumulation were different for all mentioned metabolites. Glucosides of DIBOA and its methylated counterpart DIMBOA showed decreased accumulation in both wheat genotypes. The accumulation of DIBOA di-hexoside showed a great increase in Ts, whereas it decreased in Tr. A similar trend was observed for the HDMBOA hexoside. The HMBOA glucoside showed a decreased accumulation in both genotypes.

The amino acids phenylalanine and histidine, as well as methionine and asparagine and their derivatives, were exclusive DAMs for both susceptible genotypes. We also detected indolic methoxytryptamines and gramine derivatives, in addition to *N*-acyl amines, in both time points in Hs and Ts. Relatively numerous DAMs at T2 belonged to pyrimidines. In Hs, the increase in accumulation was specifically significant for the nucleotide derivative cytidine diphosphate ribitol, whereas the accumulation of cytidine alone was highly decreased (Figure 4). Prenylated flavonoids, especially chalcone derivatives with two phenyl rings, were also observed. Derivatives of hydroxycinnamic acids, especially incorporating chlorogenic acid, constituted a large group of DAMs that were shared among susceptible varieties. Interestingly, the phytohormones hydroxyjasmonic, dihydrojasmonic, and abscisic acids were also included among the DAMs common to the susceptible genotypes.

## 4. Discussion

### 4.1. Infection Progress in Different Genotypes

Fungal growth in Bd21 was observed at a similar level to that of Hs; therefore, a close resemblance in susceptibility to *F. culmorum* between Bd and Hs can be suggested. On the other hand, Bd21 has been reported to be more resistant to *F. culmorum* than the Bd3-1 ecotype [12]. Taking into account discrepancies in Bd21′s reaction to *F. culmorum* infection, we concluded that this genotype has intermediate resistance to FHB in comparison to the selected and well-studied susceptible and resistant genotypes of barley and wheat. Many biological and histopathological studies have shown that *Fusarium* spp. infect *Brachypodium distachyon* plants in a similar manner as they infect wheat [12]. FHB’s development and strategy of cell penetration are largely conserved between barley and wheat infection, as was observed in [34]. However, wheat cultivars mostly show resistance to the initial penetration (resistance type I) [16], in contrast to barley, which shows resistance to the spread of pathogen within the head (type II) [35]. In addition to the comparison between wheat and Bd21 in terms of the response type, similarities in the response of barley and Bd were also observed in the signaling pathway, with brassinosteroid hormones participating in the *F. culmorum* recognition process [36].

In order to exclude mycotoxins from this metabolomic study, we removed the signals corresponding to mycotoxins from statistical analysis. Without this step, signals corresponding to mycotoxins would be included among the DAMs in the statistical analysis and would further influence the functional analysis. In addition to estrogenic zearalenone, we detected trichothecene mycotoxins—nivalenol, deoxynivalenol, diacetoxyscirpenol, and T2 toxin. Structurally different pathogen-derived metabolites caused slight alterations in transcription patterns in barley and wheat genotypes [37]. Extensive studies on the structural dependence of mycotoxins on plant immunity would improve our knowledge of plant–pathogen interactions in FHB.

The susceptibility of Bd to *Fusarium* species is DON-dependent [38]. The effect of ZEN on immunity of Bd in plant has not been clearly described. Moreover, the contribution of DON to the virulence of the pathogen was found to be significant in wheat, whereas the opposite result was obtained in barley [39]. The level of accumulation of pathogen-derived metabolites was highly differentiated between resistant and susceptible genotypes and it was positively correlated with pathogen biomass in plants. Similar results were previously described in [40] for wheat and in [41] for barley, where a high level of pathogen biomass was followed by a high level of mycotoxin accumulation. The profile of mycotoxin accumulation in Bd21 was different to that of other species observed in our experiments. The accumulation of trichotecenes in Bd21 was similar to that in Ts and Hs, whereas estrogenic ZEN accumulation was similar to that of resistant genotypes. The authors in [38] noted that the mycotoxins of *Fusarium* had the same phenotypic effects in Bd and wheat. The *F. culmorum* strain used in our experiments is characterized by a low level of DON and a high level of ZEN production; therefore, a low level of DON can lead to a lower level of pathogen biomass accumulation in Bd21. ZEN prevalence can affect the fungal aggressiveness in plant spikes, making the description of plant immunity even more challenging.

In addition to our comparative LC-MS analysis of mycotoxin accumulation, we quantitatively measured ZEN levels in order to check if the level of mycotoxin contamination was above the adopted safety standards. Moreover, the normalization of LC-MS signals meant that we could only obtain a picture of the kinetics and the trend of mycotoxin accumulation—one that did not reflect the real quantitative results. The ZEN level exceeded the European Union norm only in Ts at T2, but the cumulative effect of contamination could be changed over a longer period of research. Nevertheless, the level of mycotoxin accumulation was significant in all genotypes, which would consistently affect the physiological and molecular changes in plants after *F. culmorum* inoculation.

### 4.2. Antioxidant Capacity

Secondary metabolism is closely related to a plant’s antioxidative system. Antioxidant capacity and non-enzymatic antioxidants play a vital role in the detoxification of reactive oxygen species (ROS) and photosynthetic balance [42], as well as in the inhibition of mycotoxin production [43]. We observed that in resistant genotypes of barley and wheat, the antioxidant capacity was kept constant even in the advanced stage of *F. culmorum* infection, while at the same time increasing the accumulation of secondary metabolites. Compensation for antioxidative potential under FHB by increasing the low-molecular antioxidant content was previously observed in different *Poaceae* plants [44,45].

In the current experiments, high efficiency of the antioxidative system in Tr and Hr was manifested by ensuring a stable level of antioxidative capacity. On the other hand, the weakening of antioxidative activity during *F. culmorum* inoculation in susceptible Ts and Hs was accompanied by a significant increase in the *F. culmorum* biomass. *F. culmorum,* as a hemibiotroph with a short biotrophic phase, exerts nutritional benefits through its dead tissues damaged by ROS.

### 4.3. Global Analysis

Most biological studies deal with only one factor, e.g., with treatment and control groups, which can be analyzed via a one-way ANOVA using many dedicated programs, e.g., MZmine2 and MetaboAnalyst. However, in experiments such as the current case, with more than one factor, those programs do not reflect the cumulative effect of the factors. This method of analysis does not illustrate the real situation, and the results are influenced by the interaction of factors, and this brings about the risk of misleading results. To our knowledge, no other studies have employed three-factor ANOVA with the aim of detecting differentially accumulated metabolites (DAMs), equal to the use of differentially expressed genes (DEGs) in transcriptomic analysis. Previously, we analyzed metabolomic data using two-factor ANOVA with additional conditions related to the fold-change for the selection of DAMs in [46]. Moreover, by conducting a comparison based on statistically and biologically significant DAMs—rather than a simple comparison of metabolic profiles conducted separately for control and infection treatments—we greatly decreased LC-MS signal redundancy.

### 4.4. Conservative Metabolomic Response to F. culmorum among Poaceae Plants

*Fusarium*-triggered immunological responses involve massive transcriptional reprogramming within hormones, primary metabolism, and photosynthesis, as well as processes that are essential for plant growth and development [47]. Increased correlation observed in DAMs also confirmed the high level of reprogramming in the metabolome during *F. culmorum* infection. The high number of DAMs common for all studied genotypes confirmed the high level of conservativeness of the metabolomic response to *F. culmorum* infection in *Poaceae* plants, with enhanced correlation among them.

Many metabolites with altered accumulation during *F. culmorum* infection have been previously identified or annotated in *Poaceae* plants [48]. Nevertheless, their correlation and functionality remain largely unknown. Furthermore, the annotation of particular *m/z* values in relation to dedicated databases is burdened with a large error due to the use of isomeric structures and a low degree of metabolome description in *Poaceae* plants. However, assigning the entire group of compounds to a particular metabolic pathway has a higher probability of correctness than the annotation of a single *m/z* value. This gives us an opportunity for DAM localization in terms of the global plant reaction to *F. culmorum* and the grouping of metabolite classes.

We observed the robust enrichment of specific metabolic pathways related to central carbon metabolism and evolutionarily conserved phenylpropanoids, flavonoids, and porphyrin biosynthesis [49]. Carbohydrates from “Raffinose family oligosaccharides“ indicated the disturbance of sugar storage in seeds and germination processes [50]. In addition, galactose-derived oligosaccharides were predicted to be antioxidants and ABA-related signaling molecules [51]. Moreover, the monosaccharides mannose, galactose, and myo-inositol were suggested to increase FHB resistance in wheat [52] and maize [53]. Interestingly, plants from the *Poaceae* family have low levels of galactose in their cell wall polysaccharide structures; however, an additional role for galactose in the stimulation of phloem import has been suggested previously [54]. As sugar management has been shown to modulate the expression of defense-related phenylpropanoids [55], we speculate that the enrichment of these pathways is associated with a cascade of signaling, leading to reprogramming in the secondary metabolome. These speculations are supported by our observation of Bd pathway enrichment, of which sugar management seemed to be central part, as most of the Bd21-specific enriched pathways revolved around sugar metabolism. Previous studies on Bd seedlings showed that the induction of sugar metabolism was linked to osmolyte overproduction [56]. Our findings suggest a notable function for sugar metabolism in the biotic stress response in Bd plants.

The matching of T2 metabolites from the central regulatory ”2-Oxocarboxylic acids” pathway suggested disturbances in the primary metabolism of the TCA cycle and glycolysis [57]. A high score obtained for “Porphyrin and chlorophyll metabolism” at both time points corresponded to the “Heme biosynthesis” module, indicating a large disruption in photosynthesis during *F. culmorum* infection, leading to the overproduction of ROS [58], as well as a decrease in the chlorophyll content, resulting in the altered efficiency of photosynthesis. The importance of these substances in biotic stress is underlined by the fact that mycotoxins produced by necrotrophic fungi lead to the inhibition of porphyrin synthesis [59].

Phenylpropanoids in many tissues are present in a soluble form and as bonded components of the cell wall. To study the latter of these forms, an additional alkaline step in the extraction process—a method not used in our studies—should be applied. Therefore, only soluble representatives of the metabolome were considered in our analysis. Among these, different structures, such as hydroxycinnamic acids, chalcones, flavanes, flavanols, and anthocyanidins, which are broadly found in cereal spikes, were matched. The conferring of ROS scavenging or the inhibition of lipid peroxidation was related to 3, 4′-hydroxylation in the B ring of the core flavonoid skeleton, in combination with unsaturated 2-3 bonds and 4-oxo function or galloyl moiety substitution [60]. Therefore, catechin, gallocatechin from flavanols, myricetin, and quercetin (but not kaempferol from flavonols or flavones) may be efficiently involved in ROS scavenging. On the other hand, ferulic and *p*-coumaric acids, apigenin and luteolin, in addition to tyramine derivatives, have been described in the literature as fungal growth inhibitors [61]. The multitasking of hydroxycinnamic acids corroborates our observation about the drastic increase in their correlation to other metabolites under *F. culmorum* inoculation in comparison to control conditions. Among the annotated DAMs, ferulic, *p*-coumaric, caffeic, and sinapic acid, and their glycosylated forms, were central hubs, with the highest number of inter- and intra-connections in the correlation network. The possibility cannot be excluded that these metabolites contributed to the kinetics of mycotoxin accumulation observed in our experiments as inhibitors of fungal genes for trichotecene biosynthesis [62]. Such complex relationships presumably reflect the multitude of metabolic mechanisms driven by these structures.

The triggering of “Ascorbate and aldarate metabolism” as a conserved immune response may be related to ascorbic acid accumulation, as another effective antioxidant, but also as a regulator of photosynthesis and transmembrane electron transport [63].

Wheat and barley commonly showed an enriched lignan biosynthesis pathway. Monolignols precursors were previously reported to be FHB-related in resistant barley [64] and resistant wheat [65]. Pinoresinol and its derivatives were reportedly induced during FHB in the highly resistant wheat cultivar Sumai-3 [66]. Our analysis matched disturbances in pinoresinol and its derivatives in both resistant and susceptible wheat cultivars, which can be explained by the specificity of developmental stages, as well as differences in the resistance type and the *Fusarium* strains used. On the other hand, we found secoisolariciresinol and two glycosides of lignan derivatives to be commonly affected DAMs for resistant genotypes, whereas matairesinol and lyoniresinol glucoside were commonly observed DAMs for susceptible genotypes. A limited number of reports have suggested in vivo and in vitro that lignan may have an inhibiting effect on mycotoxin production caused by *Fusarium* spp. [67] and may improve cell wall thickness [68], contributing to increased plant resistance to the pathogen.

Conservative DAMs from pathways related to aromatic amino acids—“Tyrosine metabolism” at T1 and “Tryptophan metabolism“ at T2— after a stepwise of methylation could be related to the production of the antifungal alkaloids hordenine ((*N,N*-dimethyltyramine), serotonin, and tryptamine in *Poaceae* plants [69]. The induction of tyrosine- and tryptophan-derived metabolites by *Fusarium* ssp. was previously observed in Bd [5], barley [64], and wheat [62]. Tyrosine is linked to lignin formation via TAL (Tyr ammonia-lyase), an enzyme exclusively present in grasses [70].

Metabolic pathways related to unsaturated fatty acids were commonly affected both as conservative and genotype-specific factors. FHB-responsive “α-Linolenic acid metabolism” is a source of precursors for jasmonic acid and methyl jasmonate. These plant hormones are known for enhancing wheat resistance to necrotrophic pathogens [71]. The induction of JA-associated genes was also reported in wheat and Bd upon *Fusarium* infection [72].“Linoleic acid metabolism”—enriched at T2 for Bd21 in contrast to jasmonate-related α-linolenic acid—is a main source of H_2_O_2_ from the β-oxidation processes. Both of these unsaturated fatty acid precursors have been previously described in barley as antifungals [73]. Metabolites from “Arachidonic acid metabolism” are known for eliciting general stress signaling networks during FHB [74]. DAMs identified from this pathway in cooperation with lignan biosynthesis are responsible for strong changes in membrane ingredients, as well as the extracellular barrier, based on cutin and suberin [75].

### 4.5. Genotype-Specific Metabolomic Response to F. culmorum

The dominant effect of genotype and its interaction, according to ANOVA, indicated intra-species diversification in the immune response. Differences in the metabolic profiles among wheat and barley cultivars, with diversified susceptibilities to *Fusarium,* have previously been reported [64,65]. Resistance to FHB is controlled by many plant quantitative trait loci (QTL) and is highly dependent on morphological traits and environmental conditions [8,16,62]. The observed similarities in physiological features of Bd21 and susceptible genotypes of barley and wheat were in opposition to the results of the metabolic profile comparison. Bd21 showed intermediate metabolomic FHB-related changes between the metabolomes of barley and wheat.

We obtained these results with the aim of indicating differences and similarities in the immune responses to *F. culmorum* among different species and genotypes of grasses, and therefore our description of molecular biomarkers of resistance to FHB is a preliminary description and requires verification through more detailed tests.

Four-nitrogen-containing metabolites from “Caffeine metabolism” were identified in Hr as resistance-related. The presence of these particular compounds was not confirmed in our analysis based on MS/MS spectra, however, which thus represents a direction for further analyses in the search for similar four-nitrogen-containing structures which may be FHB-responsive. Matching metabolites related to purine rings from ”Caffeine biosynthesis” highlighted the triggering of nitrogen reservoirs during FHB in wheat, as was previously observed [76]. Adenosine, guanosine, and their phosphorylated forms of IMP, AMP, and GMP are critical metabolites for vegetative growth, sexual/asexual reproduction, and pathogenesis in *F. graminearum* [77]. Some transcripts encoding enzymes from purine metabolism were reported to be FHB-resistance-related in wheat [78]. Furthermore, these compounds distinguished between resistant and susceptible wheat cultivars during FHB [79]. Considering all these observations together with the fact that pyrimidines and to a lesser extent purines were highly correlated to other common DAMs, it can be generally assumed that nitrogen metabolism is related to FHB resistance in grasses.

Metabolites that were common to the resistant genotypes of wheat and barley clearly included amino acids. A consistent result was found in previous studies, indicating proline and alanine as markers of resistance to FHB in wheat [80]. Our additional database searches showed proline and serine in addition to alanine-derived panthothenic acid as factors not only for wheat but also for barley resistance. In addition, “Arginine and proline metabolism” was enriched in pathway analysis as primarily FHB-triggered in Hr. This pathway is a source of energy, as well as nitrate oxide, participating in ROS mobilization [81]. The osmolytic features of proline can be used for the protection of cellular structures [82]. Together with our study, this confirms its potential as a marker for increasing resistance in *Poaceae* plants.

Interestingly, sulfur-containing dicarboxylic acids and related compounds from the “Cysteine and methionine metabolism” pathway were also identified as resistance-conferring in Hr. Thiol-containing cysteine and glutathione are crucially important for maintaining and regulating redox homeostasis. Cysteine is the central precursor of all organic molecules containing reduced sulfur, such as phytochelatins, proteins, vitamins, cofactors, and hormones. In dicots, Brassicaceae plants, metabolites from this pathway are precursors of glucosinolate biosynthesis, which are involved in immunological reactions to herbivores and pathogens [83].

On the contrary, methionine was identified as a DAM that was shared among susceptible genotypes. Furthermore, kinetics similar to methionine were displayed for phenylalanine. This amino acid is known to have a higher level in the resistant genotype that in the susceptible genotype of barley [64]. However, in the current experiment the fold-change between control and infected plants was significant only in susceptible genotypes. Taking into account the central role of phenylalanine in secondary metabolism and its multitasking ability, its role can be profound. The same can be suggested for 12-hydroxyjasmonic acids, which were detected as DAMs common to both susceptible genotypes. In addition, 12-Hydroxyjasmonic acid is a broadly occurring inactive JA-derived metabolite. Together with its sulfated and glycosylated forms, it has been considered a switching-off element of the jasmonate signaling response in barley [84] and other plants [85]. The presence of this metabolite as susceptibility-related can presumably be related to the similar impairment of the JA-mediated response in Ts and Hs. On the contrary, JA and MeJa were annotated as conservative DAMs with a high level of correlation with other DAMs, suggesting a central role for the JA-dependent response in FHB; however, some inhibition of these processes seems to occurr in susceptible genotypes.

The accumulation of benzoxazinoids has been correlated with resistance to FHB in wheat and wild barley heads [86]. However, benzoxazinoid biosynthesis was not observed in cultivated barley and Bd. Benzoxazinoid accumulation in wheat is regulated by jasmonic acid [87]. The diverse accumulation of different benzoxazinoids was similarly observed in previous analyses, which attempted to search for a correlation between FHB resistance and benzoxazinoid accumulation [88]. These equivocal results may be related to the complex biosynthesis and regulatory pathway of benzoxazinoids, which has not yet been fully explored.

## 5. Conclusions

The combination of untargeted and targeted approaches has allowed for stupendous progress in the dissection of defense-related metabolites and in the illustration of their correlation and interaction with other elements of plant immunity. We observed that the stability of plant homeostasis during *F. culmorum* infection was driven by significant reprogramming of the metabolome. The production of antioxidants and phytoalexins was observed in ubiquitous interactions with sugar management mechanisms and photosynthesis. Plant–pathogen interaction is shaped by plant metabolites with the ability to recognize, inhibit, and counteract pathogenic microbes and their products. The genotype-specific profile of those molecules can boost or hamper the power to combat *F. culmorum*. On the other hand, pathogenic metabolites and sophisticated mechanisms of infection determine the plant’s immunity, making plant–pathogen interaction a complex process. Conserved response elements in all *Poaceae* plants use common hormone signaling pathways and evolutionarily conserved phenylpropanoids, flavonoids, purines, and pyrimidines in a highly correlated network during *F. culmorum* infection. On the other hand, specialized phytoalexins—which are limited in number in resistant genotypes—enhance the inhibition of the pathogenesis in the early stages of infection. We expect that our results concerning the conservative and genotype-specific metabolomic elements of immunity will provide information for breeders about metabolomic biomarkers of high resistance to FHB and the inhibition of mycotoxin production. The pipeline of metabolomic studies with multifactorial experiments followed by DAM selection and the elimination of signals corresponding to mycotoxins can clearly present a picture of pathogen–plant interaction at the metabolome level and may provide a baseline to deepen the search for metabolomic markers of resistance to FHB. Genotype-specific DAMs can be used in the identification of metabolic biomarkers of plant resistance to FHB, which are desired by plant breeders. On the other hand, the characterization of conserved DAMs creates an opportunity to learn about basic molecular processes related to plant immunity.

## Figures and Tables

**Figure 1 cells-11-03213-f001:**
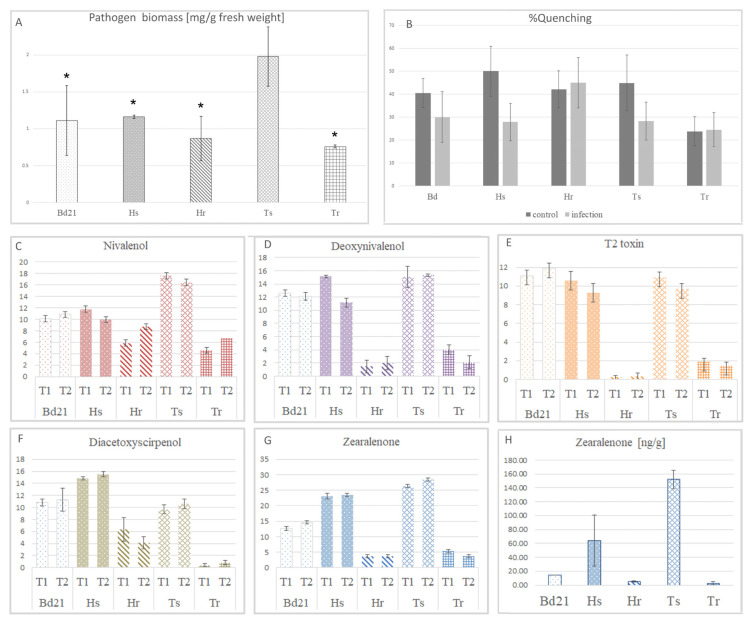
Progress of pathogenesis in different genotypes of *Poaceae* measured by (**A**) *Fusarium culmorum* biomass quantified by real-time qPCR assay. Significant differences between susceptible and resistant plants were filtered by using Student’s t-test: * *p*-value < 0.05; (**B**) DPPH radical scavenging capacities of the methanolic extracts of plants expressed as nM trolox equivalent (TE) per 100 g of extract; (**C**) LC-MS measurement of nivalenol accumulation at T1 and T2; (**D**) LC-MS measurement of deoxynivalenol accumulation at T1 and T2; (**E**) LC-MS measurement of T2 toxin accumulation at T1 and T2; (**F**) LC-MS measurement of diacetoxyscirpenol accumulation at T1 and T2; (**G**) LC-MS measurement of zearalenone accumulation at T1 and T2; (**H**) quantitative measurement of zearalenone accumulation (ng/g dry weight) in infected plants at T2; in (**C**–**H**), results for control plants were skipped as the level of mycotoxin was close to or under the detection limit; symbol “Bd” is for *Brachypodium distachyon* Bd21 line, ”Hs” is for *Hordeum vulgare* FHB—susceptible genotype, “Hr” is for *Hordeum vulgare* FHB—resistant genotype, “Ts” is for *Triticum aestivum* FHB—susceptible genotype and “Tr” is for *Triticum aestivum* FHB—resistant genotype.

**Figure 2 cells-11-03213-f002:**
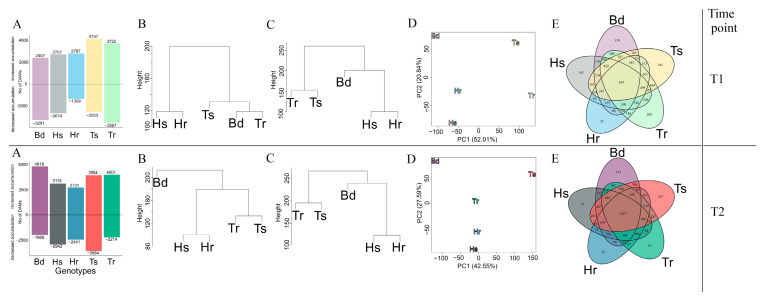
Characterization of global metabolomic changes in studied genotypes in response to *F. culmorum.* Upper panel representing results for time point T1 and bottom panel representing results for time point T2 (**A**) Bar charts presenting number of DAMs selected after ANOVA with q-value <0.05 and fold-change (FC) between inoculated and control group |log2FC| > 0.58 with increased or decreased accumulation after *F. culmorum* inoculation in each genotype, (**B**) hierarchical clustering of metabolomic profile for control plants on the basis of Euclidean distance and complete link grouping algorithm, (**C**) hierarchical clustering of metabolomic profile for group of plants infected with *F. culmorum*, (**D**) PCAs of DAMs for studied genotypes, (**E**) Venn diagrams presenting number of shared and unique DAMs for studied genotypes. Symbol “Bd” is for *Brachypodium distachyon* Bd21 line, ”Hs” is for *Hordeum vulgare* FHB—susceptible genotype, “Hr” is for *Hordeum vulgare* FHB—resistant genotype, “Ts” is for *Triticum aestivum* FHB—susceptible genotype and “Tr” is for *Triticum aestivum* FHB—resistant genotype.

**Figure 3 cells-11-03213-f003:**
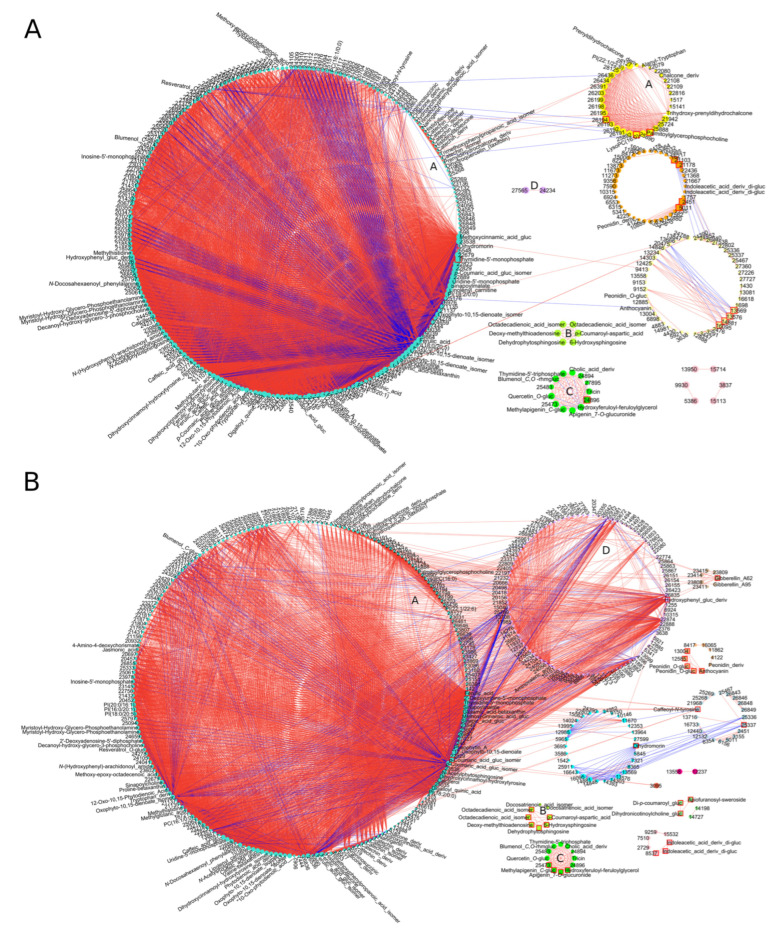
The correlation network of DAMs common for all genotypes observed (**A**) at T2 in control conditions, (**B**) at T2 after *F. culmorum* inoculation. Metabolites are represented by circles with numbers or annotated names. The size of the circle in a certain module is proportional to the number of edges/connections created by this compound and is scaled with respect to the vertex with the greatest number of edges in this module. Squares with a red border denote hubs, i.e., compounds with the highest number of connections. Edges link highly correlated DAMs. Modules of DAMs are visualized by different colors. Only edges corresponding to elements of the topological overlap matrix greater than 0.25 are shown, both within and between modules: red edges—positive correlations, blue edges—negative correlations. The modules with a large number of DAMs (>60%) occurring in both control and treatment are marked with letters (**A**–**D**). Abbreviation hex is for hexoside, deriv is for derivative.

**Figure 4 cells-11-03213-f004:**
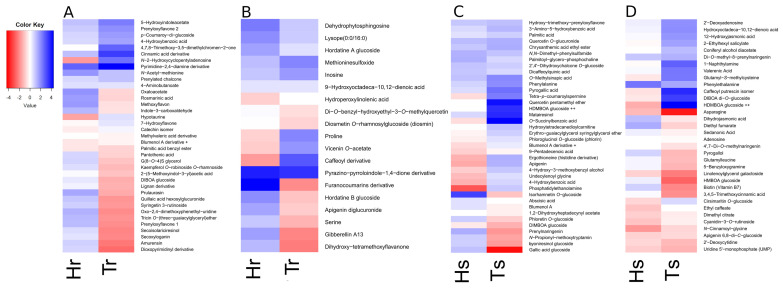
Differences in accumulation of selected DAMs shared for FHB-resistant Hr and Tr genotypes: (**A**) at T1 time point, (**B**) at T2 time point or FHB-susceptible Hs and Ts genotypes (**C**) at T1 time point and (**D**) at T2 time point. Details of annotation are available in Appendix A. Symbol ”Hs” is for *Hordeum vulgare* FHB- susceptible genotype, “Hr” is for *Hordeum vulgare* FHB-resistant genotype, “Ts” is for *Triticum aestivum* FHB-susceptible genotype and “Tr” is for *Triticum aestivum* FHB-resistant genotype. The color key refers to differences in abundance (shown in log_2_ scale) between infected and control plants. The rows were initially sorted in a descending order, first by Hr and Hs, respectively, then by Tr and Ts, and then secondly reordered based on row means. The symbols “+” and “++” represent two the same metabolites present at different treatment types and time points.

**Table 1 cells-11-03213-t001:** Enrichment analysis of LC-MS signals annotated to KEGG Metabolic pathways and structural groups selected as the most enriched for common DAMs. Only pathways with FDR of matching <0.3 and pathway impact I > 0.5 were presented in table as statistically significant. Entire functional enrichment results and compounds annotation can be found in Appendix A. Only structural classes enriched with FDR of matching <0.01 were presented in table as statistically significant. Entire structural enrichment results can be found in Appendix A.

Genotype	Time Point T1		Time Point T2	
Enrichment	Functional	Structural	Functional	Structural
common	Galactose metabolism	Benzamides	Galactose metabolism	Benzamides
	Porphyrin and chlorophyll metabolism	Monosaccharides	Ascorbate and aldarate metabolism	Purines
	Flavonoid biosynthesis	Amino acids and peptides	Porphyrin and chlorophyll metabolism	Amino acids and peptides
	Phenylpropanoid biosynthesis	Purines	Arachidonic acid metabolism	Monosaccharides
	Tyrosine metabolism	Porphyrins	2-Oxocarboxylic acid metabolism	Isoprenoids
	Isoquinoline alkaloid biosynthesis	TCA acids	Tryptophan metabolism	Indoles
		Pyrimidines	Phenylpropanoid biosynthesis	Porphyrins
		Cinnamic acids		Pyrimidines
		Sphingoid bases		Glycosyl compounds
		Pyridoxamines		Cinnamic acids
		Benzenes		TCA acids
		Fatty acids and conjugates		Sphingoid bases
		Short-chain acids and derivatives		Benzenediols
		Isoprenoids		Tryptamines
		Disaccharides		Disaccharides
		Benzenediols		Eicosanoids
				Imidazoles
				Phenols
				Organooxygen compounds
				Aldehydes
Hs	Biosynthesis of secondary metabolites—other antibiotics	Benzamides	Biosynthesis of secondary metabolites—other antibiotics	Monosaccharides
	Galactose metabolism	Amino acids and peptides	Diterpenoid biosynthesis	Disaccharides
	Ascorbate and aldarate metabolism	Porphyrins	alpha-Linolenic acid metabolism	Purines
	alpha-Linolenic acid metabolism	TCA acids	Arachidonic acid metabolism	
	Diterpenoid biosynthesis	Monosaccharides		
		Isoprenoids		
Hr	Arginine and proline metabolism	Amino acids and peptides	Caffeine metabolism	Purines
	2-Oxocarboxylic acid metabolism	Benzamides	2-Oxocarboxylic acid metabolism	
	Phenylpropanoid biosynthesis		Purine metabolism	
Ts	Biosynthesis of secondary metabolites—other antibiotics	Purines	Arachidonic acid metabolism	Benzamides
	Arachidonic acid metabolism	Benzamides	Diterpenoid biosynthesis	Amino acids and peptides
	Caffeine metabolism	Monosaccharides		Monosaccharides
	Amino sugar and nucleotide sugar metabolism	TCA acids		Quinones and hydroquinones
		Imidazoles		Sterols
		Amino acids and peptides		Cyclic alcohols
		Indoles		Tryptamines
		Eicosanoids		Isoprenoids
				Benzoic acids
				Pyrimidines
				Fatty acids and conjugates
				Purines
				Cinnamic acids
				Eicosanoids
Tr	Flavonoid biosynthesis	Benzamides	Biosynthesis of secondary metabolites—other antibiotics	Benzamides
	Flavone and flavonol biosynthesis	Purines	Cutin, suberin, and wax biosynthesis	
	Arachidonic acid metabolism	Monosaccharides	Linoleic acid metabolism	
	Phenylpropanoid biosynthesis	Imidazoles	Biosynthesis of unsaturated fatty acids	
		Cinnamic acids	Arachidonic acid metabolism	
		Isoprenoids	Galactose metabolism	
		Eicosanoids	alpha-Linolenic acid metabolism	
		Flavonoids		
Bd21	Amino sugar and nucleotide sugar metabolism	Monosaccharides	Phenylpropanoid biosynthesis	Purines
	Glycolysis/gluconeogenesis	Benzamides	Purine metabolism	Pyrimidines
	alpha-Linolenic acid metabolism	TCA acids	Linoleic acid metabolism	Phenylpropanoids
	Pentose phosphate pathway	Purines	Flavone and flavonol biosynthesis	
	Fructose and mannose metabolism	Phosphate esters		
	Citrate cycle (TCA cycle)			
	Galactose metabolism			
	Inositol phosphate metabolism			
	Glyoxylate and dicarboxylate metabolism			
	Carbon fixation in photosynthetic organisms			

## Data Availability

Datasets from positive and negative ionization are deposited in a publicly available database Mendeley Data.

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
