# Peer review of "Metabolomic Aspects of Conservative and Resistance-Related Elements of Response to Fusarium culmorum in the Grass Family"

_cells, 2022, doi:10.3390/cells11203213_

Round 1
Reviewer 1 Report
In this study, combined targeted and untargeted metabolomics was used provides comprehensive knowledge about significant elements of plant immunity with potential of being molecular biomarkers of enhance resistance to FHB in grass family. Thorough examination of Bd21 metabolome in juxtaposition with barley and wheat diversified genotypes facilitate their setting as model grass for plant-microbe interaction.
However, the manuscript is too long. It is difficult to get the main conclusion from the manuscript. What is the important molecular biomarkers related with resistance to FHB? In addition, authors did not verify the biomarkers use direct experiment data.
In this study, the fungus Fusarium culmorum KF846 was inoculated, please provide the infromation for toxins production of this fungus. In Line 135-142.
For detect the biomass of Fusarium culmorum, which method was used. qPCR? Please provide the primers for qPCR.
In the full manuscript, the "in vitro" should be italic.
ZEA or ZEN? please use only one name in the whole manuscript.
For antioxidant and mycotoxin analysis samples were collected at 6 days after inoculation. I think 6 days is too short for mycotoxins production. More days should be designed.
Figure 2,Figure3 and Figure 5 are not clear. Please replace it use clear figure.
In Line 49: Fusarium should be italic.
Line 84: It should be B. distachyon consider it appeared for the second time.
Line 135: It should be F. culmorum consider it appeared not for the first time.
Line 142: 2 ×106 spores / ml., please correct it in the manuscript.
Line 176-177: please correct the error.
Line 337: What is the UE norm?
Author Response
Dear Reviewer,
We are very grateful to the Reviewer for a great effort in exposing the shortcomings and errors of primary version of our manuscript entitled “Metabolomic aspects of conservative and resistance-related elements of response to Fusarium culmorum in the grass family”. Please find enclosed our revised version in which we applied referees’ suggestions. We trust that proposed changes will be satisfying Reviewers as well as enriches and completes our manuscript. All the comments we received on this study have been taken into account in improving the quality of the article, and we present our reply to each of them separately. We hope that these changes to the manuscript will facilitate the decision to publish this study in journal. In any case, we are open to consideration of any further comment on our answers.
In this study, combined targeted and untargeted metabolomics was used provides comprehensive knowledge about significant elements of plant immunity with potential of being molecular biomarkers of enhance resistance to FHB in grass family. Thorough examination of Bd21 metabolome in juxtaposition with barley and wheat diversified genotypes facilitate their setting as model grass for plant-microbe interaction.
However, the manuscript is too long. It is difficult to get the main conclusion from the manuscript.
We also very much appreciate of suggestions, which have been very helpful in improving the manuscript. The authors admit that it is very difficult for them to give up any results. We have reduced some parts of the results as well as the discussion. We hope that the Result and Discussion section is now easier to follow. We deleted Table 1 and moved it to supplementary table S2 as additional sheet “Summary”. Results describing the 3-way ANOVA were also removed. We also removed part of discussion corresponding to mentioned results. We removed Figure 4. and results describing the conservative DAMs with the highest diversification among genotypes. We have focused just only on DAMs specific for resistant or susceptible genotypes from section 3.9.1 presented on Figure 5 (after correction Figure with number 4).
We also deleted some part of discussion related to functional analysis. We regrouped discussion subsections and limited its number from 8 to 4.
In order to limit the discussion subtitles we removed the subtitles:
4.2. “Mycotoxins accumulation in spikes” and incorporate it to section 4.1 related to pathogen infection.
4.2 “Statistical analysis- added values” and incorporated it to subsection Global analysis.
4.6. “Functional metabolomics” and incorporate it to section 4.4 Conservative metabolomic response to F. culmorum among Poaceae plant.
We hope the changes are acceptable and increase clarity and quality of the manuscript.
What is the important molecular biomarkers related with resistance to FHB?
We are very grateful for pointing out that this information should be highlighted. The biomarkers are indicated in Figure 5 and described in results in actually section 3.9. Comparison for predominant DAMs for resistant and susceptible genotypes of barley and wheat aims at indication on possible metabolomic biomarkers of increasing of resistance to Fusarium pathogens. The title of this results subsection was changed to “Putative metabolomic biomarkers of resistance to F. culmorum belong to amino acids, pyrimidines, phenolics and jasmonic acid derivatives” for betters clarification that it is related to biomarkers selection.
In addition, authors did not verify the biomarkers use direct experiment data.
We indeed have not verify this results, however as the reviewers pointed out, the number of results is very extensive and detailed. Our experiments were aimed at describing the pathogen-plant interaction and at identifying the differences and similarities in the immune response at the metabolomic level between different species of Poaceae. Verification for presented metabolomics biomarkers will certainly be the next step of work. Verification should be comprehend and detailed analysis of targeted metabolomics with MS/MS spectra for selected MS signals identification. But it should certainly be extended by gene expression analysis, studies on mutant impaired in production of particular metabolites, and/or external implementation of particular metabolites on plants inoculated with F. culmorum followed by morphological observation the plant immune response. The given examples of verification methods should certainly be extended with further detailed analyses from various fields of molecular biology and physiology, the results of which will make up a separate publication being an excellent continuation of the current manuscript. In order to clearly indicate that these are comparative studies and relate to preliminary results, we added in the discussion the information about the need to conduct additional analyses to verify the obtained results. To the title of subsection 3.9 the word “Putative” was added.
In this study, the fungus Fusarium culmorum KF846 was inoculated, please provide the infromation for toxins production of this fungus. In Line 135-142.
Additional characterisation of used isolate of F. culmorum was added in indicated lines.
For detect the biomass of Fusarium culmorum, which method was used. qPCR? Please provide the primers for qPCR.
The analysis was done by qPCR as was described in section 2.2. Thank you for indication missing data for primers for qPCR. It was added to supplementary methods 1
In the full manuscript, the "in vitro" should be italic.
We are grateful for careful study current manuscript and indication of this mistake. It was corrected
ZEA or ZEN? please use only one name in the whole manuscript.
We change incorrect abbreviation “ZEA” on currently used for zearalenone “ZEN”
For antioxidant and mycotoxin analysis samples were collected at 6 days after inoculation. I think 6 days is too short for mycotoxins production. More days should be designed.
We planned our experiment with special care on the basis of suggestion of expert in the field of plant-pathogen interaction Measurement on 6th day was performed correspondingly to metabolites measurement which was also at that day. As was presented in the current manuscript the 6th day was sufficient to detected different mycotoxins production (Figure 1 B-H).
Mycotoxins are pathogen-derived metabolites which start to be produced after penetrating the host tissues very quickly after inoculation and even shorter days after inoculation was previously taken for LC-MS molecular analysis as was reported in many publications (e.g. Pasquet et al. 2014, Blümke et al. 2015). Additional, Peraldi et al. 2011 noticed that mycelial growth of Fusarium graminearum and F. culmorum was detectable on the host surface – Brachypodium distachyon from between 12 and 36 hpi. However, on the other hand, it should be highlighted that the host plant does not necessary show the infection symptoms despite the pathogen's. However, we conducted analysis of expression of pathogen’s genes responsible for mycotoxins biosynthesis (mentioned in Supplementary method 1) of inoculated plants also on 6th day. This data was not shown in manuscript as only analysis for barley genotypes were completed. It served as preliminary data for planning the main experiment. The level of expression of Tri5, Tri6 and ZEA2 gene was observed at 6th day, therefore it was taken for analysis.
On the other hand, for in vitro analysis of Fusarium proliferatum treated by plant host extract, Górna et al. reported that fumonisin levels in liquid media changed during 14 d of culturing and a massive increase of FBs amounts was usually noticed in the culture media shortly after the extracts (pineapple,, asparagus, maize, garli extract) were added (on 5th and 6th day).
Górna, K., Pawłowicz, I., Waśkiewicz, A. and Stępień, Ł. (2016) Fusarium proliferatum Strains Change Fumonisin Biosynthesis and Accumulation When Exposed to Host Plant Extracts. Fungal Biology, 120, 884-893. https://doi.org/10.1016/j.funbio.2016.04.004
Blümke A, Sode B, Ellinger D, Voigt CA. Reduced susceptibility to Fusarium head blight in Brachypodium distachyon through priming with the Fusarium mycotoxin deoxynivalenol. Mol Plant Pathol. 2015 Jun;16(5):472-83. doi: 10.1111/mpp.12203. Epub 2014 Oct 22. PMID: 25202860; PMCID: PMC6638442.
Peraldi, A., Beccari, G., Steed, A. et al. Brachypodium distachyon: a new pathosystem to study Fusarium head blight and other Fusarium diseases of wheat. BMC Plant Biol 11, 100 (2011). https://doi.org/10.1186/1471-2229-11-100
Pasquet, J.C.; Chaouch, S.; Macadré, C.; Balzergue, S.; Huguet, S.; Martin-Magniette, M.L.; Bellvert, F.; Deguercy, X.; Thareau, V.; Heintz, D.; et al. Differential gene expression and metabolomic analyses of Brachypodium distachyon infected by deoxynivalenol producing and non-producing strains of Fusarium graminearum. BMC Genomics 2014, 15, 629, doi:10.1186/1471-2164-15-629.
Figure 2,Figure3 and Figure 5 are not clear. Please replace it use clear figure.
Figures with better resolution were sent separately (not in text) .Tiffs format which is adequate to Editor requirements. Adding the figures to .docx file greatly lowers the resolution. However, the proper figures were sent and its clearly presented.
In Line 49: Fusarium should be italic.
We are grateful for careful study of current manuscript and indication of this mistake. It was corrected
Line 84: It should be B. distachyon consider it appeared for the second time.
It was corrected
Line 135: It should be F. culmorum consider it appeared not for the first time.
It was corrected
Line 142: 2 ×106 spores / ml., please correct it in the manuscript.
Yes, it was mistake. It should be 106. It was corrected.
Line 176-177: please correct the error.
It was corrected.
Line 337: What is the UE norm?
It referred to norms applied for European Union. The abbreviation was extended.

Reviewer 2 Report
The paper by Anna Piasecka et al. reported conservative and resistance-related metabolomic responses to Fusarium culmorum in the grass family. This is a very interesting paper that would be helpful for the reseachers in the field of FHB research. The authors performed a compreshensive analysis of metabolites from the dimensions of different species, different genotypes within a species, control vs F.C. inoculation, and different timepoints after inoculation. I am particulally interested in the conservative metabolic pathways in the grass family, which did enrich our understanding of host-F.C.interactions. In general, the paper is well-written and organized, and the conclusion is convincing. I would suggest the authors carefully check and correct the grammatical or spelling errors thoroughly. I list some of them, but not all, as the followings:
(1) ’shown’ almost everywhere
(2) P3 L142: 2 ×106 spores / ml
(3) P4 L167: purchase
(4) P7L304: ‘Resistant genotypes impaired mycotoxin production by F. culmorum’ is unclear;
(5) P7L306: od
(6) P9L371-372: not clear
(7) p18l633: ‘then’ should be ‘than’?
(8) P19L646: Prior to exclude...
(9) P19L696: systemin
(10) P24L952: marker of enhance resistance
(11) ......
Author Response
Dear Reviewer,
We are very grateful to the Reviewer for a great effort in exposing the shortcomings and errors of primary version of our manuscript entitled “Metabolomic aspects of conservative and resistance-related elements of response to Fusarium culmorum in the grass family”. Please find enclosed our revised version in which we applied referees’ suggestions. We trust that proposed changes will be satisfying Reviewers as well as enriches and completes our manuscript. All the comments we received on this study have been taken into account in improving the quality of the article, and we present our reply to each of them separately. We hope that these changes to the manuscript will facilitate the decision to publish this study in journal. In any case, we are open to consideration of any further comment on our answers.
The paper by Anna Piasecka et al. reported conservative and resistance-related metabolomic responses to Fusarium culmorum in the grass family. This is a very interesting paper that would be helpful for the reseachers in the field of FHB research. The authors performed a compreshensive analysis of metabolites from the dimensions of different species, different genotypes within a species, control vs F.C. inoculation, and different timepoints after inoculation. I am particulally interested in the conservative metabolic pathways in the grass family, which did enrich our understanding of host-F.C.interactions. In general, the paper is well-written and organized, and the conclusion is convincing. I would suggest the authors carefully check and correct the grammatical or spelling errors thoroughly.
We are grateful for careful study current manuscript and indication of grammatical and spelling errors. We additionally used the help of a native speaker from Language Editing Services of MDPI platform to improve quality of manuscript reading. We hope that now the current manuscript is much more pleasant to read
I list some of them, but not all, as the followings:
(1) ’shown’ almost everywhere
We change the word mainly with word “showed “, but in some phrases e.g. L 550 sentence ‘was shown “ replaced with “was indicated”, in L 323 replaced with “was displayed” and in L 750 replaced with “revealed”
(2) P3 L142: 2 ×106 spores / ml
Yes, it was mistake. It should be 106. It was corrected.
(3) P4 L167: purchase
The wording was changed to “was purchased”
(4) P7L304: ‘Resistant genotypes impaired mycotoxin production by F. culmorum’ is unclear;
The subtitled was changed to “Production of pathogen-derived mycotoxins was impaired in resistant genotypes of Poaceae”
(5) P7L306: od
The wording was changed to “of”
(6) P9L371-372: not clear
Thank you for pointing out this imprecise sentence. It should indicate on differences in number of DAMs among genotypes. It was replaced with: “Proportionally lesser DAMs both with decreased as well as increased accumulation were observed for barley genotypes whereas wheat and Bd had extended number of DAMs”. For clarification we changed also the next sentence; “In wheat and Bd the differences in DAMs number between T1 and T2 was more evident, especially for DAMs with decreased accumulation.”
(7) p18l633: ‘then’ should be ‘than’?
Indeed, it was a mistake, thank you for indicating this.
(8) P19L646: Prior to exclude...
The wording was changed to: “In order to..”
(9) P19L696: systemin
The wording was changed to: “system in”
(10) P24L952: marker of enhance resistance
The wording was changed to: “…marker of resistance”
(11) ......
Reviewer 3 Report
The manuscript 'Metabolomic aspects of conservative and resistance-related elements of response to Fusarium culmorum in the grass family' could be ineteresting paper as FHB is very dangerous disease and any new approach to combat this disease is desirable.
However, I think that results and discussion is to long and that it should be shorten, and only main results shown.
Also, manuscript has many technical and grammatical mistakes.
Here are just some, but whole manuscript should be reviwed and corrected:
pg1. line 41 Cereal crops in Europe are (not is)
line 43- should you move this internet page in reference?
line 46 -bracket with number 1, should be non-italic
Pg2. line 61-Poaceae- italic
line 63 Did you metntioned somewhere earlier about abbreviations Bd or Bx?
line 67- Poaceae-italic
line 95- Poaceae-italic
Pg3. line 114 was selected
lines 140-141 why you used PDA medium in inoculum production, as this medium is used for detection of Fusarium species for coloration? Usually, inoculum is produced at different wayes (SNA medium, bubble breeding etc).
line142- 2X106 spores/ ml- should it 10 on 6 be
line 143- why the inoculations was done at the milk stage? Usually it should occur at flowering.
Pg 6. line 291 where you measured FHB disease severity?
Is ti possible to calculate severity based on biomass of fungi? I think not.
Results and discussion are to long, can you elucidate the most important results and adapta discussion to that?
Like this is very difficult to follow, with so many data inside.
Page 18. line 643 What is F.c.?
line 649- ill? or will
lines 666-668 can you please the characterize F. culmorum strain in Material and methods? Whici isolate have you been using etc?
line 688- Poaceae italic
Line 690- Fusarium-italic
line 696- In the current research, not our
line 696- please seperate system and in
In general there are to many subtitles in the discussion.
Pg. 20 line 737- you have extra bracket
line 740- Poaceae italic
line 743- number reference in bracket are not correctly writen, (53-54), not (53), (54)
line 749- the same as previous
line 762-please eject 'for review see'
Author Response
Dear Reviewer,
We are very grateful to the Reviewer for a great effort in exposing the shortcomings and errors of primary version of our manuscript entitled “Metabolomic aspects of conservative and resistance-related elements of response to Fusarium culmorum in the grass family”. Please find enclosed our revised version in which we applied referees’ suggestions. We trust that proposed changes will be satisfying Reviewers as well as enriches and completes our manuscript. All the comments we received on this study have been taken into account in improving the quality of the article, and we present our reply to each of them separately. We hope that these changes to the manuscript will facilitate the decision to publish this study in journal. In any case, we are open to consideration of any further comment on our answers.
The manuscript 'Metabolomic aspects of conservative and resistance-related elements of response to Fusarium culmorum in the grass family' could be ineteresting paper as FHB is very dangerous disease and any new approach to combat this disease is desirable.
However, I think that results and discussion is to long and that it should be shorten, and only main results shown.
Authors admit that it is very difficult for them to give up any results. We have reduced some parts of the results as well as the discussion (addition explanation below). We hope that the Result and Discussion section is now easier to follow.
Also, manuscript has many technical and grammatical mistakes.
We are grateful for careful study current manuscript and indication of grammatical and spelling errors. We additionally used the help of a native speaker from Language Editing Services of MDPI platform to improve quality of manuscript reading.
Here are just some, but whole manuscript should be reviwed and corrected:
pg1. line 41 Cereal crops in Europe are (not is)
We fully agree that this word should be changed.
line 43- should you move this internet page in reference?
The internet link was moved to section References
line 46 -bracket with number 1, should be non-italic
Thank you for pointing out this imprecision. It was corrected
Pg2. line 61-Poaceae- italic
line 67- Poaceae-italic
line 95- Poaceae-italic
All was corrected
line 63 Did you metntioned somewhere earlier about abbreviations Bd or Bx?
Indeed, abbreviation “Bx” and “Bd“ was not mentioned before in main text. We changed the abbreviation Bx to sentence: “biosynthetic genes of benzoxazinoids” and the abbreviation Bd to Brachypodium distachyon
Pg3. line 114 was selected
The wording was changed
lines 140-141 why you used PDA medium in inoculum production, as this medium is used for detection of Fusarium species for coloration? Usually, inoculum is produced at different wayes (SNA medium, bubble breeding etc).
The sporulation was obtained on PDA medium that is why it was used for inoculum production.
line142- 2X106 spores/ ml- should it 10 on 6 be
Yes, it was mistake. It should be 106. It was corrected.
line 143- why the inoculations was done at the milk stage? Usually it should occur at flowering.
Thank you for pointing out this mistake, we are very sorry for them. It was indeed flowering stage taken for inoculation as was indicated properly at BBCH scale However, the wrong name of the stage was described. “the milk stage of the spikes” was changes on “flowering stage”.
Pg 6. line 291 where you measured FHB disease severity? Is ti possible to calculate severity based on biomass of fungi? I think not.
We are very grateful for pointing out the wrong phrase. We used “severity” for illustration of progress of pathogen biomass production in inoculated plants. Pathogen biomass measurement was the main source of infection progress monitoring in our experiments. Indeed this is the case, the “severity” definition is rather related to disease symptoms referred to as virulence. Therefore, we changed the subtitle “FHB disease severity was diversified among genotypes” to “Progress of F. culmorum infection was diversified among genotypes”.
Results and discussion are to long, can you elucidate the most important results and adapta discussion to that? Like this is very difficult to follow, with so many data inside.
We also very much appreciate of suggestions, which have been very helpful in improving the manuscript. The authors admit that it is very difficult for them to give up any results. We have reduced some parts of the results as well as the discussion. We hope that the Result and Discussion section is now easier to follow. We deleted Table 1 and moved it to supplementary table S2 as additional sheet “Summary”. Results describing the 3-way ANOVA was also removed. We also removed part of discussion corresponding to mentioned results. We removed Figure 4. and results describing the conservative DAMs with the highest diversification among genotypes. We have focused just only on DAMs specific for resistant or susceptible genotypes from section 3.9.1 presented on Figure 5 (after correction Figure with number 4).
We also deleted some part of discussion related to functional analysis. We regrouped discussion subsections and limited its number from 8 to 4.
In order to limit the discussion subtitles we removed the subtitles:
4.2. “Mycotoxins accumulation in spikes” and incorporate it to section 4.1 related to pathogen infection.
4.2 “Statistical analysis- added values” and incorporated it to subsection Global analysis.
4.6. “Functional metabolomics” and incorporate it to section 4.4 Conservative metabolomic response to F.culmorum among Poaceae plant.
We hope the changes are acceptable and increase clarity and quality of the manuscript.
Page 18. line 643 What is F.c.?
F.c is related to an abbreviation for Fusarium culmorum. However, we haven’t used it in manuscript before, therefore we changed it to name used throughout the manuscript F. culmorum.
line 649- ill? or will
Thank you for indicating this mistake, it was changed to “will”.
lines 666-668 can you please the characterize F. culmorum strain in Material and methods? Whici isolate have you been using etc?
Characterization of F. culmorum strain used in current experiments was added in 2.2 section of Materials and Methods
line 688- Poaceae italic
Line 690- Fusarium-italic
We are grateful for careful study current manuscript and indication of this mistake. All mistakes for Poaceae and Fusarium was corrected.
line 696- In the current research, not our
The word “our” was changed to “current”.
line 696- please seperate system and in
It was changed.
In general there are to many subtitles in the discussion.
The correction was described above.
Pg. 20 line 737- you have extra bracket
Thank you for indication of this mistake, it was deleted.
line 740- Poaceae italic
It was corrected.
line 743- number reference in bracket are not correctly writen, (53-54), not (53), (54)
line 749- the same as previous
Thank you for indication of these mistakes, all were corrected.
line 762-please eject 'for review see'
The correction was done.
Round 2
Reviewer 1 Report
The manuscript has been significantly improved according to the comments of the reviewers. Now, it is acceptable in the present form.
Author Response
Dear Reviewer,
Thank you very much for taking the time to read our research and expressing positive opinions about them and for the inappreciable comments that made our manuscript more valuable and understandable for readers.
Sincerely yours,
Anna Piasecka
Aneta Sawikowska,
Natalia Witaszak
Agnieszka Waśkiewicz
Marta Kańczurzewska
Joanna Kaczmarek
Justyna Lalak-Kańczugowska
Reviewer 3 Report
I do not have any additional comments.
Author Response

(The authors gave the same response as above.)
